A taxonomically detailed and large-scale view of the factors affecting the distribution and abundance of tree species planted in private gardens of Christchurch city, New Zealand

Quan Wei 1
http://orcid.org/0000-0001-6169-6660 Sullivan Jon J. 1 Jon.Sullivan@lincoln.ac.nz
Meurk Colin D. 2
http://orcid.org/0000-0002-8462-6160 Stewart Glenn H. 3
1 Department of Pest-Management and Conservation, Lincoln University , Lincoln, Canterbury , New Zealand
2 Manaaki-Whenua, Landcare Research , Lincoln, Canterbury , New Zealand
3 Department of Environmental Management, Faculty of Environment, Society and Design, Lincoln University , Lincoln, Canterbury , New Zealand
Culham Alastair
Electronic publication date: 2021 Mar 26
Publication date: 2021
Volume: 9
Electronic Location ID: e10588
Received 2019 Oct 15; Accepted 2020 Nov 25
Copyright: © 2021 Quan et al.
Copyright year: 2021
Copyright holder: Quan et al.
License: This is an open access article distributed under the terms of the Creative Commons Attribution License, which permits unrestricted use, distribution, reproduction and adaptation in any medium and for any purpose provided that it is properly attributed. For attribution, the original author(s), title, publication source (PeerJ) and either DOI or URL of the article must be cited.
License URL: https://creativecommons.org/licenses/by/4.0/

Keywords: Urban ecology, Urban trees, Private gardens, Planting choices

Funding: The authors received no funding for this work.

==============================
A city’s planted trees, the great majority of which are in private gardens, play a fundamental role in shaping a city’s wild ecology, ecosystem functioning, and ecosystem services. However, studying tree diversity across a city’s many thousands of separate private gardens is logistically challenging. After the disastrous 2010–2011 earthquakes in Christchurch, New Zealand, over 7,000 homes were abandoned and a botanical survey of these gardens was contracted by the Government’s Canterbury Earthquake Recovery Authority (CERA) prior to buildings being demolished. This unprecedented access to private gardens across the 443.9 hectares ‘Residential Red Zone’ area of eastern Christchurch is a unique opportunity to explore the composition of trees in private gardens across a large area of a New Zealand city. We analysed these survey data to describe the effects of housing age, socio-economics, human population density, and general soil quality, on tree abundance, species richness, and the proportion of indigenous and exotic species. We found that while most of the tree species were exotic, about half of the individual trees were local native species. There is an increasing realisation of the native tree species values among Christchurch citizens and gardens in more recent areas of housing had a higher proportion of smaller/younger native trees. However, the same sites had proportionately more exotic trees, by species and individuals, amongst their larger planted trees than older areas of housing. The majority of the species, and individuals, of the larger (≥10 cm DBH) trees planted in gardens still tend to be exotic species. In newer suburbs, gardens in wealthy areas had more native trees than gardens from poorer areas, while in older suburbs, poorer areas had more native big trees than wealthy areas. In combination, these describe, in detail unparalleled for at least in New Zealand, how the tree infrastructure of the city varies in space and time. This lays the groundwork for better understanding of how wildlife distribution and abundance, wild plant regeneration, and ecosystem services, are affected by the city’s trees.

Introduction

Planted trees form a natural foundation for urban ecosystems (Lawton, 2007). The density, age, health, species traits, and spatial arrangement of a city’s trees all play important roles in determining a city’s wild biology, ecosystem functioning, and ecosystem services (Rowntree & Nowak, 1991; McPherson & Rowntree, 1993; Lawton, 2007). For example, the composition of a city’s wild animal and fungal communities will be affected by the shelter and food provided by a city’s trees (Beatley, 2011). Wild plant regeneration in a city’s wild places is also strongly influenced by the trees planted around these wild places, and which of these produce viable pollen and seeds (Whelan et al., 2006; Sullivan et al., 2009; Doody et al., 2010; Overdyck & Clarkson, 2012). Increasing research focus is also being placed on the importance of urban trees for human health and wellbeing (Attwell, 2000; Ulrich, 1984; Ulrich et al., 1991; Fraser & Kenney, 2000).

It has been suggested that using native plants in different urban habitats can make contribution to attract and conserve wildlife in cities (McKinney, 2002; Tallamy, 2007). These suggestions came from the historical co-evolution of native plants and native insects, which will help native insects feed or reproduce on native plants more effectively (Comba et al., 1999). Several studies demonstrated the correlations between native plants and native insects and birds such as Lepidoptera larvae (Burghardt, Tallamy & Gregory Shriver, 2009; Tallamy & Shropshire, 2009), pollen and nectar feeding bees (Hopwood, 2008) and native insectivorous and frugivorous birds (Burghardt, Tallamy & Gregory Shriver, 2009) to support the conclusion that enhancing the biomass and diversity of native plants will increase the diversity and abundance of insects, which will affect on bird communities (Tallamy, 2004; Burghardt, Tallamy & Gregory Shriver, 2009).

As urban areas increase globally, private gardens play an increasingly important role in some countries as they can potentially make contributions to urban biodiversity (Sawyer, 2005; Smith et al., 2005; Loram et al., 2007; Stewart et al., 2009), ecosystem functioning (Sperling & Lortie, 2010) and providing habitats for native wildlife (Cameron, 2012). Private gardens are common in urban areas and comprise a substantial proportion of the urban area (Gaston et al., 2005; Van Heezik et al., 2013). The estimated proportions of private garden area in cities range from 16% in Stockholm, Sweden (Colding, Lundberg & Folke, 2006), through to around 25% in UK (Loram et al., 2007) and 36% in Dunedin, New Zealand (Mathieu, Freeman & Aryal, 2007). Private gardens are therefore a large proportion of all urban green space of urban area, such as 35% in Edinburgh and 47% in Leicester (Loram et al., 2007). Considering that private gardens are probably the biggest single contributor to urban green space (Gaston et al., 2005), they may also be the largest source of planted trees (Smith et al., 2006).

Many environmental, and non-environmental factors can potential influence the make-up of a city’s trees. Human activities substantially alter the soil in urban environment (Scharenbroch, Lloyd & Johnson-Maynard, 2005) and this can affect planted trees (Jim, 1998). A number of non-environmental factors also influence planting choices in private gardens (Shaw, Miller & Wescott, 2017; Van Heezik et al., 2013), including social patterns (Caldicott, 1997), marketing influences (Shaw, Miller & Wescott, 2017), environmental knowledge (Head & Muir, 2005), and economic conditions (Daniels & Kirkpatrick, 2006). Vegetation composition and structure can also be related to householder socio-economic status, as well as residents’ motivations and attitudes (Hope et al., 2003; Martin, Warren & Kinzig, 2004; Van Heezik et al., 2013). Several studies have explored environmental attitudes on gardens and planting (Head & Muir, 2004, 2005; Zagorski, Kirkpatrick & Stratford, 2004; Lohr & Pearson-Mims, 2005). One study showed a strong relationship between gardeners’ values and the species composition of their gardens, with the gardeners who have pro-environmental views more likely to have more native plants in their gardens (Zagorski, Kirkpatrick & Stratford, 2004).

Understanding the ecology of a city’s nature, and ecosystem services, requires a detailed knowledge of the city’s planted trees, and that means documenting the trees in private gardens. The logistics of negotiating access onto thousands of different private properties makes it difficult to study the tree and shrub composition of private urban gardens in high spatial and taxonomic detail across large areas of cities. Knowledge of city tree scapes is therefore often limited to smaller spatial scales (Van Heezik et al., 2013), or to what can be learned from studying street side trees (Mulvaney, 2001), or registered notable trees (Wyse et al., 2015). Larger spatial scale analyses can be achieved from aerial and satellite imagery but with that comes reduced taxonomic resolution (Clarkson, Wehi & Brabyn, 2007; Mathieu, Freeman & Aryal, 2007).

Through unfortunate events, Christchurch city in New Zealand was able to provide a large-scale, multi-suburb, taxonomically detailed look at the trees planted in the city’s private gardens. On 4 September 2010, the city was shaken by a 7.1 magnitude earthquake centred 50 km west of the city, and, over the next year, thousands of subsequent quakes, including a shallow and deadly magnitude 6.3 quake directly under the city on 22 February 2011 (Bradley & Cubrinovski, 2011; Morgenroth & Armstrong, 2012; Harding & Jellyman, 2015). The considerable damage to properties and infrastructure led to large areas of the city, including more than 7,000 homes, being purchased by the central New Zealand government, the largest contiguous area being 443.9 hectares of the city’s eastern suburbs.

The Canterbury Earthquake Recovery Authority (CERA) was established to manage the demolition and rebuild of the damaged parts of the city (Vallance & Tait, 2013). CERA contracted a botanical survey of all the established garden trees in what became known as Christchurch’s ‘Residential Red Zone’. This survey informed CERA’s subsequent management of the area and care was taken during building demolition to leave as many established garden trees as possible.

The large damaged areas of Christchurch city offered an unusual large-scale and detailed opportunity to examine the tree and shrub composition of a city’s private gardens. What areas of the city have the highest density and diversity of trees and shrubs? To what extent is this affected by housing age, socio-economics, human population density, and general soil quality? Which factors affect the proportion of indigenous and exotic tree species planted in private gardens?

Specifically, we address the following questions.What is the composition of residential garden trees in eastern Christchurch?

Do younger suburbs have higher native tree abundance and species richness than older suburbs?

Do social factors (human population density and economic deprivation) affect tree abundance and richness, and the proportion of native to exotic trees?

Does soil versatility (a measure of soil suitability for crop cultivation) have a positive effect on native tree abundance and richness?

Methods

Study sites

Christchurch is the third largest city in New Zealand, with a resident population of over 400,000 people. Internationally, Christchurch is a young city, founded in 1850 (Wilson, 1989). It is a coastal city located on the relatively dry eastern side of the South Island, mostly built on a mosaic of shingle lobes deposited by the Waimakariri River to the north, interspersed and overlaid with swamplands, waterways, and sandhills (Wilson et al., 2005). There are a range of natural habitats within the built city, including wetlands, coastal habitats, grasslands, drylands, hills and one small remnant of old growth forest (Christchurch City Council, 2000). The climate is cool temperate and oceanic (McGann, 1983). Christchurch has a relatively low mean annual rainfall of around 660 mm although rain falls all year round, often interspersed in the summer months with hot and desiccating foehn winds (McGann, 1983) (Fig. 1).

Figure 1 Map showing the location of Christchurch (left top) in New Zealand and the location of the Residential Red Zone in Christchurch.

An ID card was used as a pass to go to the Residential Red Zone Area. This card was given by CERA (The Canterbury Earthquake Recovery Authority), the public service department of New Zealand charged with coordinating the rebuild of Christchurch and the surrounding areas following the 22 February 2011 earthquake. The Residential Red Zone was a public exclusion zone created on 23rd June 2011 in eastern Christchurch after the 2010–2011 earthquakes. All houses in the most damaged areas were directed to be removed by CERA and, as much as possible, the garden trees were saved. The remaining vegetation includes most of the larger ornamental trees planted in the private gardens and in adjacent parks. Our research focuses on the 14 suburbs along the Avon River in eastern Christchurch that make up the largest contiguous area of the residential red zone, at 443.90 hectares (Fig. 2).

Figure 2 Map showing the 14 suburbs in Residential Red Zone area.

Data sources

Tree map

Tree inventory data from the Residential Red Zone were provided by CERA as GIS files. This map contains 27,698 mapped trees and large shrubs, identified to species (or, in some cases, genus) from the >7,000 private properties acquired by CERA. Some Diameter at Breast Height (DBH) values were unrealistically big, or small, indicating some data entry errors. Unrealistic data values were filtered and the data with DBH values between 5 cm and 2 m, inclusive, were used in the analysis. Most of the DBH data (18,925, 97%) were used in this research.

We split the tree inventory dataset into a 100 m × 100 m square grid across the study site, so that we could standardise the scale of this and other variables for our analysis.

Population data

The human population data came from the 2013 Census from Statistics NZ (https://stats.govt.nz) (Statistics New Zealand, 2013). For this factor, a grid layer of 100 m × 100 m was applied to standardise the scale of analyses in QGIS. When more than one census meshblock overlapped a grid cell, an average value was calculated proportional to the area that each meshblock occupied in the grid square. Figure 3A shows the distribution of population densities across the study area.

Figure 3 Maps showing human population, economic deprivation, soil versatility and maximum age of suburban housing in all 100 m × 100 m grid squares across the Residential Red Zone study area in eastern Christchurch.

(A) Estimated resident human population per grid cell, (B) Mean economic deprivation per grid cell, (C) Mean soil versatility per grid cell, (D) Maximum age of suburban housing per grid cell.

Economic deprivation data

Economic deprivation data came from the New Zealand Index of Socio-economic Deprivation for Individuals (NZiDep) which was made in 2013 (Atkinson, Salmond & Crampton, 2014). This index is applied to the same meshblocks as the population census data. NZDep2013 deprivation scale is from 1 to 10 in which 1 is least deprived and 10 is most deprived. As with the human population values, economic deprivation values were applied to a 100 m × 100 m square grid across the study site. Figure 3B shows the distribution of deprivation values in the study area.

Soil data

Soil data was obtained from the soil map of Christchurch City from the NZ Soil Survey Report 16 held by Manaaki Whenua-Landcare Research (Webb, Smith & Trangmar, 2006). We used the soil versatility rating as a measure of the overall quality of the soil conditions. The definition of versatility here is the ability of land to support the production and management of a range of crop plants on a sustained yield basis and is mainly assessed in terms of physical soil characteristics (Webb, Smith & Trangmar, 2006). It assumes that nutrient and soil moisture limitations are overcome by fertiliser application and irrigation (Webb, Smith & Trangmar, 2006), as will likely be the case in private gardens. This data set uses five ranked soil versatility classes, from Class 1 (very high versatility) through to Class 5 (very low versatility). Figure 3C maps the range of soil versatility values across the study area.

Housing maximum age

To assess the range of garden ages in the study area, all the grids were assigned the year in which houses were first built. This was extracted manually for every 100 m × 100 m grid square from the historical aerial photography layers available on the Canterbury Maps website (https://canterburymaps.govt.nz). The time ranges available in the aerial photography were from 1940–2010 excluding 1950–1954. The earliest year in which more than three houses were established was used as the maximum garden age for each grid. This avoided the bias created by single old farm houses that were present in rural parts of the city prior to suburban house subdivisions being built. Figure 3D maps the range of housing ages across the study area.

Plant nomenclature

In general, plant names were made consistent with Nga¯ Tipu o Aotearoa, the New Zealand Plant Names Database (https://nzflora.landcareresearch.co.nz). A group of plants (766 individuals, 2.77%) could not be identified by the surveyors and were named ‘other sp’ in the database. Because native plants in gardens could be reliably identified by local botanical contractors, it is assumed that the biostatus of unidentified plants was ‘exotic’ for the analysis. To analyse the data, all unknown species recorded as ‘Other sp’ were conservatively treated as one species.

Plant biostatus

New Zealand-wide plant biostatus data came from the New Zealand Organisms Register (http://www.nzor.org.nz). All the trees categorised as ‘Native’ or ‘Exotic’ were then further split into local native categories: ‘Native to Christchurch’ (examples: Pittosporum eugenioides, Pittosporum tenuifolium and Plagianthus regius) or ‘Non-native to Christchurch’ (J. Sullivan and Colin D. Meurk, 2014–2019, personal observation. Examples: Pittosporum crassifolium, Hoheria populnea and Corynocarpus laevigatus), and ‘Naturalised’ (examples: Acer pseudoplatanus, Quercus robur and Alnus glutinosa) or non-naturalised cultivated ‘Exotic’ (examples: Rhododendron sp. and Camellia sp.) (Mahon, 2007; Gatehouse, 2008). For trees only identified to genus (1656, 8.48%) where the genus contained no native species, biostatus was ‘Exotic’. Similarly, if the genus only contained native species, the biostatus was conservatively assigned to ‘Non-native to Christchurch’ (For trees which are native to Christchurch, they will be identified to species.) Where the genus contains species that are found in other countries as well as New Zealand, but in which 75% of the species, wild or cultivated, are known to be in NZ are native (based on the New Zealand Plants Biosecurity Index Version: 2.0.0, 2014 Ministry of Agriculture and Forestry), they were recorded as ‘Non-native to Christchurch’. Otherwise they were recorded as ‘Exotic’.

Analysis

Package ‘AICcmodavg’ (Mazerolle, 2017) was used in R (R Core Team, 2017) to compare plausible generalised linear models (GLMs) involving the factors human population, economic deprivation, soil versatility, age of suburban housing, and total tree number (all measured per 100 m × 100 m grid square). This package includes functions to implement model selection and multi-model inferences based on Akaike’s Information Criterion (AIC) and the second-order AIC (AICc) (Akaike, 1973). If the difference between AICc values were ≥2, the model with smaller AICc value was considered the best model (Arnold, 2010).

Several models were compared in the analyses of both native and exotic big (DBH ≥ 10 cm) and small (DBH < 10 cm) trees, such as an all three interactions model, all two-order interactions model, two-order interactions without human population, two-order interactions without versatility model, two-order interactions without established year model, and a no interactions model. There are more than one best model, so another package ‘MuMIn’ was applied in R (R Core Team, 2017) to average the best models. This package averages models based on model weights derived from AICc.

For plotting model predictions, near minimum and maximum values of each factor were selected. For human population, minimum population was 0 and maximum was 36 per grid cell. For tree number, it was 7–21 trees per grid cell, for soil versatility 3–5, and for economic deprivation, 3–7. The age of oldest suburban housing was plotted for 1940 and 2000.

Results

Species composition

There were 413 identified taxa (species or genus) recorded in the 14 suburbs of the Christchurch Residential Red Zone area. Exotic plants (naturalised species and exotics only cultivated) made up 80.6% (333) of taxa, while only about 11% were native to Christchurch (Fig. 4). However, for the individual trees, over half of them were native, mostly trees were native to Christchurch (Fig. 4).

Figure 4 Percentage of tree species (A) and individual trees (B) with different biostatus in the Christchurch Residential Red Zone area.

Of the different suburbs, Avonloop had the most trees mapped per hectare (21.8 trees per hectare, including 7.9 native trees and 13.9 exotic trees), New Brighton followed with 19.5 trees per hectare and third was Linwood with 17.8 trees per hectare. Dallington, Avondale, Bexley, Burwood and Travis suburbs, all of which contain areas of relatively recent housing subdivisions, had a low TPH which were all under 4 trees mapped per hectare (Fig. 5; Table 1).

Figure 5 Maps showing big exotic trees, big native trees, small exotic trees and small native trees in the Christchurch residential red zone area, using a 100 m × 100 m grid.

(A) Density of big exotic trees per grid cell (DBH 10 cm), (B) Big native trees per grid cell, (C) Small exotic trees per grid cell (DBH < 10 cm), (D) Small native trees per grid cell.

Table 1 Total tree number/tree densities of the residential red zone areas in different Christchurch suburbs. TPH is trees per hectare.

Suburb	Area (ha)	Native/TPH	Exotic/TPH	Total/TPH	
Linwood	2.8	14/4.9	36/12.8	50/17.8	
Richmond North	8.3	30/3.6	78/9.4	108/13	
Aranui	14.4	40/2.8	68/4.7	108/7.5	
Wainoni	7.8	30/3.8	65/8.3	95/12.1	
Richmond South	18.7	38/2	92/4.9	130/7	
Avonloop	3.7	29/7.9	51/13.9	80/21.8	
Avonside	50	54/1	153/3	207/4.1	
Dallington	62.2	62/1	158/2.5	220 3.5	
New Brighton	2.7	20/7.3	33/12.1	53/19.5	
Avondale	57.3	50/0.9	149/2.6	199/3.4	
Rawhiti	35.3	45/1.3	115/3.3	160/4.5	
Bexley	52.6	49/0.9	116/2.2	165/3.1	
Burwood	71.3	67/0.9	195/2.7	262/3.7	
Travis	56.1	56/1	127/2.3	183/3.3	

Planting changes in the residential red zone

DBH (Diameter at breast height) of exotic and native trees

Comparing the DBH distributions of all exotic and all native trees showed that trees with large DBH were more likely to be exotic (Fig. 6). The DBH of most native trees was under 50 cm. There were 13% more native trees than exotic trees for trees whose DBH was under 30 cm. In contrast, for trees with DBH over 30 cm, there were 7.8% more exotic than native ones. This suggests that native trees in these gardens have smaller stature as adults than the exotics, and/or that a higher proportion them are of more recently planted (they are younger than the exotics).

Figure 6 The DBH (Diameter at breast height) distribution of all (A) exotic and (B) native trees.

Only DBH values between 5 cm and 2 m are included.

Distribution of native species and native individual trees

Overall, the proportion of plant species that were native changed little regardless of housing age (Fig. 7). In the oldest areas of the city, natives made up 55% of the garden tree species. In the most recently established suburbs, this was 60%, an insignificant difference.

Figure 7 Percentage of tree species and individuals that were native, plotted against the decade in which each 100 m by 100 m grid square was first developed for suburban housing.

(A) The percentage of native species, and (B) the percentage of all individual native trees. Neither relationship is statistically significant.

The same result was seen for individual trees. Overall, there were no big changes from 1940s to 2000s. The percentage of individual native trees in 2000 was still was under 60% (Fig. 7).

Different sizes of native trees

Greater differences were seen when the trees were divided into large trees (DBH ≥ 10 cm) and small trees (DBH < 10 cm). The percentage of big native tree species dropped from ca. 50% to ca. 40% from older to younger subdivisions (Fig. 8). The percentage of big individual native trees showed a similar tend (Fig. 8).

Figure 8 (A & C) The percentage of different sized native trees and (B & D) the percentage of different sized native individual trees.

In comparison, for small trees, the percentage of both native species richness and individual trees increased about 10% in the past 60 years.

Environmental factors affecting garden tree composition

Tables 2 and 3 show the results of the generalised linear models assessing the combined effects of human population density, soil versatility, economic deprivation, housing age, and total tree density on the number of native and exotic trees per 100 m × 100 m grid square. Large trees (DBH ≥ 10 cm) and small trees (DBH < 10 cm) were analysed separately. All factors were included in some or all of the best models (within 2 AICc values of the best fitting model).

Table 2 Model Selection for big trees (DBH 10 cm).

Pop = Estimated human population in grid in 2013, Estab = Subdivision establishment year (first year with housing development per grid), Dep = Mean economic deprivation from the New Zealand Index of Socioeconomic Deprivation for Individuals in 2013, Vers = Mean soil Versatility, Total = Total number of trees in grid. The table shows the model parameters for the 17 models compared. The binomial response variable was the total number of individual native and exotic trees.

No.	(Intercept)	Pop	Estab	Dep	Vers	Totaltrees	Pop:Estab	Pop:Dep	Pop:Vers	Pop:Total	
1	63.3146279	0.02808691	−0.0330635	−7.7873588	0.30054103	−0.968914	–	–	–	−0.0013111	
2	59.6948347	0.02548032	−0.031291	−7.2648	0.30478016	−0.9004362	–	–	–	−0.0011968	
3	59.4627635	0.44493699	−0.0310599	−7.631372	0.28356915	−0.9366715	−0.000212	–	–	−0.0013735	
4	56.8464619	0.02744566	−0.029757	−7.5553328	1.60723539	−0.9565062	–	–	–	−0.0013102	
5	62.9109287	0.03348604	−0.0328476	−7.6798677	0.29838176	−0.9724266	–	–	−0.0014477	−0.0013471	
6	63.6905248	0.03071133	−0.0332708	−7.828571	0.30413823	−0.9713035	–	−0.0004894	–	−0.0013032	
7	56.8782613	0.3646779	−0.0298163	−7.1832044	0.29062445	−0.8802423	−0.0001724	–	–	−0.0012575	
8	51.8774045	0.02461164	−0.027298	−6.9681225	1.85267825	−0.8826984	–	–	–	−0.0011909	
9	57.670053	0.85906952	−0.0301877	−7.6915227	0.2855962	−0.9176291	−0.0004157	−0.0024982	–	−0.001393	
10	59.4926156	0.02987725	−0.0311775	−7.1964537	0.30292342	−0.9054667	–	–	−0.0011551	−0.0012294	
11	60.1188125	0.02889267	−0.0315293	−7.3088892	0.30959755	−0.9022065	–	−0.0006457	–	−0.0011841	
12	51.2497022	0.48248401	−0.0268582	−7.3355547	1.86947287	−0.9185266	−0.0002315	–	–	−0.0013781	
13	63.919885	0.05090548	−0.0334126	−7.7109065	0.31018642	−0.9865493	–	−0.0019697	−0.0032887	−0.0013608	
14	59.5042158	0.43887688	−0.0310812	−7.6293316	0.28373586	−0.9372978	−0.0002088	–	−5.90E-05	−0.0013741	
15	56.5268971	0.03278121	−0.0295843	−7.4521762	1.58924694	−0.9601076	–	–	−0.0014294	−0.0013457	
16	57.3180217	0.02983878	−0.0300118	−7.5973633	1.58414067	−0.958934	–	−0.0004438	–	−0.0013031	
17	55.2163206	0.7677857	−0.0290072	−7.2541496	0.29235047	−0.8634141	−0.0003707	−0.0024166	–	−0.0012799	
Estab:Dep	Estab:Vers	Estab:Total	Dep:Vers	Dep:Total	Vers:Total	df	logLik	AICc	delta	weight	
0.00405167	–	0.00050819	−0.0348023	–	−0.0058152	11	−1,145.8603	2,314.20942	0	0.18312268	
0.00379817	–	0.00047652	−0.0340247	−0.0012121	−0.0061268	12	−1,145.2897	2,315.15818	0.94875194	0.11395235	
0.00396457	–	0.00049184	−0.0319959	–	−0.0057741	12	−1,145.5601	2,315.69913	1.48970102	0.08694762	
0.00393477	−0.0006665	0.00050195	−0.0355281	–	−0.0058983	12	−1,145.7444	2,316.0677	1.85827088	0.07231423	
0.00399152	–	0.00051025	−0.0326501	–	−0.0059097	12	−1,145.7464	2,316.07167	1.86225019	0.07217049	
0.0040761	–	0.00050936	−0.0356593	–	−0.0058008	12	−1,145.8332	2,316.24526	2.03583243	0.06617086	
0.00374928	–	0.00046601	−0.0318126	−0.0011062	−0.0060663	13	−1,145.0956	2,316.86782	2.6583968	0.04847062	
0.00364911	−0.0007894	0.00046771	−0.0348536	−0.0012634	−0.006238	13	−1,145.1282	2,316.93305	2.72362531	0.04691529	
0.0040054	–	0.00048197	−0.0336692	–	−0.0056604	13	−1,145.1332	2,316.94306	2.73363346	0.04668111	
0.00375863	–	0.00047919	−0.0323349	−0.0011719	−0.0061917	13	−1,145.2178	2,317.11223	2.90280164	0.04289501	
0.00382541	–	0.00047742	−0.0351392	−0.0012359	−0.0061135	13	−1,145.2428	2,317.16213	2.95271015	0.04183784	
0.00381475	−0.0008097	0.0004827	−0.0326195	–	−0.0058711	13	−1,145.3917	2,317.46004	3.25061448	0.03604792	
0.00401432	–	0.00051759	−0.0333735	–	−0.0059721	13	−1,145.4926	2,317.66169	3.45226177	0.03259065	
0.00396343	–	0.00049217	−0.0319503	–	−0.0057785	13	−1,145.56	2,317.79661	3.58718925	0.03046449	
0.00387688	−0.0006584	0.00050405	−0.033395	–	−0.0059905	13	−1,145.6334	2,317.94336	3.73393967	0.02830919	
0.00395927	−0.000653	0.00050314	−0.0362909	–	−0.0058835	13	−1,145.7222	2,318.12101	3.91158247	0.02590316	
0.00379489	–	0.00045719	−0.0334352	−0.0010733	−0.0059469	14	−1,144.6967	2,318.17553	3.96610906	0.0252065	

Table 3 Model Selection for small trees (DBH < 10 cm).

Pop = Estimated human population in grid in 2013, Estab = Subdivision establishment year (first year with housing development per grid), Dep = Mean economic deprivation from the New Zealand Index of Socioeconomic Deprivation for Individuals in 2013, Vers = Mean soil Versatility,Total = Total number of trees in grid. The table shows the model parameters for the 22 models compared. The binomial response variable was the total number of individual native and exotic trees.

No.	(Intercept)	Pop	Estab	Dep	Vers	Total trees	Pop:Estab	Pop:Dep	Pop:Vers	Pop:Total	
1	−3.2376608	−1.346392944	0.001872311	–	–	0.01960859	0.00068351	–	–	–	
2	−3.4878808	−1.542969062	0.001951612	0.018514269	–	0.0201027	0.00078323	–	–	–	
3	−3.1494251	−1.3175469	0.001848282	–	–	0.01717368	0.00066581	–	–	0.00039363	
4	5.12075594	−1.454621433	−0.002423261	−1.958297259	–	0.02020707	0.00073961	–	–	–	
5	−1.8896779	−1.382708055	0.001164929	–	0.01084209	0.01950888	0.00070221	–	–	–	
6	−5.5521165	−1.326817545	0.003048823	–	–	0.13469083	0.00067365	–	–	–	
7	−3.396077	−1.514950675	0.001925747	0.018621467	–	0.01763974	0.0007659	–	–	0.00039885	
8	6.35253016	−1.40776984	−0.003022383	−2.212669156	–	0.01704185	0.00071198	–	–	0.00051575	
9	−1.8350241	−1.598019809	0.001081605	0.019460301	0.013394377	0.02000364	0.0008115	–	–	–	
10	−5.4442482	−1.523510437	0.002946841	0.01822069	–	0.11769886	0.00077342	–	–	–	
11	−3.7705315	−1.432845511	0.0020831	0.023400897	–	0.02004648	0.00072958	−0.0008435	–	–	
12	−3.4910325	−1.548381283	0.001937339	0.025237361	–	0.02171796	0.00078587	–	–	–	
13	−1.9042599	−1.35174435	0.001194314	–	0.010032079	0.01714133	0.00068348	–	–	0.0003841	
14	−5.5815456	−1.295402027	0.003084863	–	–	0.13773179	0.00065461	–	–	0.00039802	
15	5.60665727	−1.490075227	−0.002687	−1.856179737	0.007506551	0.02014609	0.00075773	–	–	–	
16	3.34464984	−1.438236105	−0.00151974	−1.945893849	–	0.10614133	0.00073135	–	–	–	
17	5.0874383	−1.459663223	−0.002420001	−1.944926315	–	0.02159814	0.00074208	–	–	–	
18	4.90915497	−1.412763712	−0.002320413	−1.933527552	–	0.02018438	0.00071921	−0.000326	–	–	
19	−4.2482841	−1.363314439	0.002363377	–	0.011085533	0.13826579	0.00069245	–	–	–	
20	0.07589424	−1.400522402	0.000155906	–	−0.42640131	0.01949183	0.00071136	–	–	–	
21	−1.7282096	−1.410248393	0.001079999	–	0.012290886	0.01952362	0.00071673	–	−0.0002916	–	
22	−1.8935623	−1.382009772	0.001165488	–	0.011553222	0.0196436	0.00070186	–	–	–	
Estab:Dep	Estab:Vers	Estab:Total	Dep:Vers	Dep:Total	Vers:Total	df	logLik	AICc	delta	weight	
–	–	–	–	–	–	5	−1,102.8224	2,215.74652	0	0.15588725	
–	–	–	–	–	–	6	−1,102.3029	2,216.7485	1.00198142	0.09445677	
–	–	–	–	–	–	6	−1,102.541	2,217.22467	1.47815179	0.07444474	
0.00100431	–	–	–	–	–	7	−1,101.5655	2,217.32157	1.57505205	0.07092386	
–	–	–	–	–	–	6	−1,102.727	2,217.59655	1.85002484	0.06181342	
–	–	−5.85E−05	–	–	–	6	−1,102.7596	2,217.66177	1.91525134	0.05983001	
–	–	–	–	–	–	7	−1,102.0155	2,218.22143	2.47490957	0.04522627	
0.0011336	–	–	–	–	–	8	−1,101.1009	2,218.4471	2.70057824	0.04040057	
–	–	–	–	–	–	7	−1,102.1585	2,218.50748	2.76096051	0.03919906	
–	–	−4.96E-05	–	–	–	7	−1,102.2579	2,218.70623	2.95971226	0.03549092	
–	–	–	–	–	–	7	−1,102.2669	2,218.72432	2.9777969	0.03517145	
–	–	–	–	−0.000347383	–	7	−1,102.2801	2,218.75068	3.00415966	0.03471088	
–	–	–	–	–	–	7	−1,102.4594	2,219.10923	3.36270637	0.02901404	
–	–	−6.13E−05	–	–	–	7	−1,102.4722	2,219.13485	3.38832464	0.02864476	
0.00095269	–	–	–	–	–	8	−1,101.5222	2,219.28979	3.54326602	0.02650942	
0.00099787	–	−4.37E−05	–	–	–	8	−1,101.5304	2,219.30621	3.55968552	0.02629267	
0.00100046	–	–	–	−0.000299654	–	8	−1,101.5485	2,219.34224	3.5957202	0.02582319	
0.00099268	–	–	–	–	–	8	−1,101.5603	2,219.36593	3.61940972	0.02551912	
–	–	−6.04E−05	–	–	–	7	−1,102.6599	2,219.51032	3.76380208	0.02374168	
–	0.00022419	–	–	–	–	7	−1,102.7162	2,219.62293	3.87640911	0.02244187	
–	–	–	–	–	–	7	−1,102.7247	2,219.63987	3.8933511	0.02225257	
–	–	–	–	–	−3.51E−05	7	−1,102.7268	2,219.64411	3.89758769	0.02220548	

Proportion of small native trees

The proportion of small native trees increased from old to new suburbs in both low- and high-deprivation areas (Fig. 9). For small native trees, human population density is an important factor which increased their percentage especially in younger subdivisions. For economic deprivation, at least in the last 40 years, high-deprivation areas had a higher proportion of small native trees than low deprivation areas.

Figure 9 (A–I) The generalised linear models’ predicted effects of different levels of factors on the proportion of small native trees (DBH < 10 cm) in 100 m × 100 m grid squares.

Total trees was of less interest than other four factors so it was set as an average value which is 14 throughout.

When the value for resident human population density is 0, the sites can be treated as public parks or reserves. These areas had the highest percentage of small native trees compared with areas with higher population density (meaning more private gardens). The percentage of small native trees in these areas of public parks/reserves was lower in recently established low deprivation areas than recent higher deprivation areas.

Proportion of big native trees

For oldest housing age, higher human population areas had a higher proportion of big native trees than low population areas (Fig. 10). As population density increased, the proportion of big native trees in high deprivation areas started to decrease. As soil versatility increased, the proportion of big native trees also increased in low deprivation area.

Figure 10 (A–I) The generalised linear models’ predicted effects of different levels of factors on the proportion of big native trees (DBH 10 cm) in 100 m × 100 m grid squares.

Total trees was of less interest than other four factors so it was set as an average value which is 14 throughout.

Overall there was a decline in the proportion of big trees that were native in recent subdivisions. This decline was most pronounced in low deprivation areas.

Discussion

The Christchurch residential red zone tree survey reveals substantial spatial, and temporal, variation in the structure of the city’s tree scape in private gardens (Figs. 5, 9 and 10). Some of this structure has the potential to influence the city’s wild biology, ecosystem functioning, and ecosystem services. For example, some areas of the city have more native trees planted in private gardens than others, influenced particularly by an area’s age, human population density, and affluence. Native trees differ from most exotic trees in Christchurch city by producing bird-dispersed fleshy fruits (Burrows, 1994), and being hosts to a diversity of native herbivorous insects (Spiller & Wise, 1982). As such, we anticipate that these areas of the city with higher densities of native trees in private gardens will be more suitable for native birds (Day, 1995; Van Heezik, Smyth & Mathieu, 2008).

About 80% of the tree species in Christchurch’s private gardens were exotics and only 20% native to New Zealand (Fig. 4). For comparison, (Loram et al., 2008) describe the proportion of all species (not just trees) in gardens in the United Kingdom, and report similar to higher proportions of exotic species in private gardens, with 32% of native species in gardens in Belfast, 29% in Cardiff, 30% in Edinburgh, 29% in Leicester and 29% in Oxford. In Auckland city, the percentage of garden trees that are native is around 25% (McDonnell, Hahs & Breuste, 2009).

While Christchurch’s tree flora is heavily dominated by exotic species, native trees were on average planted much more frequently, so much so that around 55% of all planted trees were native to New Zealand, and the great majority of those were native to the wider Christchurch area (Fig. 4). This suggests a big difference between the trees of private gardens and the City Council planted trees of Christchurch’s urban public green spaces. Big native tree species are not common in Christchurch’s public green spaces (Stewart et al., 2004; Stewart et al., 2009), and >80% of planted street and park trees are exotic (Stewart et al., 2004). The great majority of native trees in Christchurch will therefore be in the city’s private gardens.

In Christchurch, the proportion of big trees that were native was less in recent housing subdivisions than older areas of housing, in contrast to the smaller trees. This could be because there are many more choices of exotic garden plants and nurseries are in the business of trying to find new plant fashions that attract buyers. Most native tree species sold in plant nurseries are small/young plants, and, compared with native trees species, exotic trees tend to be bigger and more expensive, so younger areas of housing and wealthier areas would be expected to have more big exotic trees initially planted. Native trees like Pittosporum tenuifolium, Griselinia litoralis, and Olearia paniculata tend to instead be purchased for use in hedging in new subdivisions.

As in many parts of the world, the recent history of Christchurch has included a growing appreciation of the values of native species (Stewart et al., 2004). This can be reflected in planting choices both in public parks and private gardens. It is notable that there were proportionately more small native trees in the gardens in areas of a high population density. While the tree survey data did not explicitly separate trees from parks from trees from private gardens (and the great majority of the area was private gardens), we can use the resident human population density effect to estimate how tree planting differs in public park areas from private gardens. Our model predictions with resident human population is set to zero (Figs. 9 and 10) can be interpreted as the equivalent of public parks and reserves. When this is done, it is interesting that the proportion of small trees in public parks was unaffected by suburb age, while for private gardens, the oldest suburbs had fewer small native trees than public parks, while the newer suburbs had many more small native trees than public parks (Fig. 9). This suggests to us that changing public perception of the values of native trees is being reflected by a more rapid change in planting choices in private gardens than public spaces.

Generally speaking, the proportion of small trees that are native increases in younger suburbs. That suggests that more people, both gardeners and landscape architects, are realising the importance of native trees in our urban ecosystem (or that they require generally less effort to maintain). Doody et al. (2010) found 54% of surveyed Christchurch residents in the suburb of Riccarton would like to plant native species from a local urban forest in their gardens and (Van Heezik et al., 2013) found in Dunedin about 40% of garden holders in their research have a preference for planting native species in their gardens. However, in Australia, almost 90% of the respondents indicated they would like to plant native plants in their garden in the future, and the most preferred garden type was a lawn with native plants from the six choices (Shaw, Miller & Wescott, 2017). That raised another question: why don’t they plant more native plants in their garden currently (Shaw, Miller & Wescott, 2017)? It was found that the relationship between having an intention to plant native plants and planting native plants is not straight-forward (Kollmuss & Agyeman, 2002).

Economic deprivation was an important social factor correlated with native garden trees but in complex ways. We expected exotic trees to be more abundant in the wealthier areas. One reason is that tree species sold in plant nurseries are expensive, and a diversity of garden plants is not affordable for poorer people (Bigirimana et al., 2012). However, the reality turns out to be different. It can be found in the prediction of big and small native tree proportions. In recently established areas, wealthier areas typically have more native tree species than poor areas, whereas in older areas of the city, wealthier areas have similar or often fewer native trees than poorer areas. This may signal a changing attitude towards native trees in private gardens, with wealthier people now being more likely than in the past to invest in native trees when establishing their gardens. Several studies have shown a positive association between wealth of suburbs and vegetation biodiversity, in USA (Hope et al., 2003; Kinzig et al., 2005) and in Australia (Luck, Smallbone & O’Brien, 2009) as well as New Zealand (Wyse et al., 2015). Our results suggest that this is not always a simple relationship.

Conclusion

Private gardens are an important kind of urban green space, holding much of a city’s tree diversity. In the case of Christchurch city, the great majority of native trees in the city are planted in private gardens. The private choices being made by Christchurch residents are therefore likely to make a big difference to the city’s ecology. Further work is now required to assess the extent to which the patterns in trees planted in Christchurch private gardens are affecting the city’s wildlife and ecosystem services.

For more recently established housing developments, gardens in more affluent areas had more native trees than less affluent areas. However, this differed in older suburbs, where gardens in less affluent areas had more native big trees than gardens in more affluent areas. This is an encouraging sign that Christchurch residents are placing more value on having native trees in their neighbourhoods.

Our results are consistent with an increasing realisation among Christchurch citizens of the values of native tree species, as they are planting more native trees in their gardens. However, even if there are more native species in urban gardens than before, the percentage of all tree species that are native remains low (<20%). The number and diversity of exotic trees being planted has increased alongside increases in native trees planting. About a quarter of trees planted in Christchurch gardens are now exotic species that have naturalised in New Zealand and are capable of regenerating wild in the city as woody weeds. The tree planting choices being made in the city’s private gardens can have positive, and negative, effects on the wider environment.

Supplemental Information

Supplemental Information 1 Tree data exported from Tree map of whole Red Zone.

Click here for additional data file.

Supplemental Information 2 Maps with data used in QGIS and R code.

Click here for additional data file.

Special thanks go to the Canterbury Earthquake Recovery Authority and Treetech Specialist Treecare Ltd. for collecting such a valuable urban tree dataset during such a stressful time for Christchurch city.

Additional Information and Declarations

Competing Interests

Author Contributions

Field Study Permissions

Data Availability

The authors declare that they have no competing interests.

Wei Quan performed the experiments, analysed the data, prepared figures and/or tables, authored or reviewed drafts of the paper, and approved the final draft.

Jon J. Sullivan conceived and designed the experiments, analysed the data, prepared figures and/or tables, authored or reviewed drafts of the paper, and approved the final draft.

Colin D. Meurk analysed the data, authored or reviewed drafts of the paper, data and methods and editing, and approved the final draft.

Glenn H. Stewart conceived and designed the experiments, analysed the data, authored or reviewed drafts of the paper, data and methods, and approved the final draft.

The following information was supplied relating to field study approvals (i.e., approving body and any reference numbers):

An ID card was used as a pass to go to the Residential Red Zone Area. This card was given by CERA (The Canterbury Earthquake Recovery Authority), the public service department of New Zealand charged with coordinating the rebuild of Christchurch and the surrounding areas following the 22 February 2011 earthquake.

The following information was supplied regarding data availability:

The raw data is available at Figshare:

Quan, Wei (2019): Chapter2 data. figshare. Dataset. DOI 10.6084/m9.figshare.9976106.v1.

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
