# Peer review of "A taxonomically detailed and large-scale view of the factors affecting the distribution and abundance of tree species planted in private gardens of Christchurch city, New Zealand"

_PeerJ, doi:10.7717/peerj.10588_

## Round 0.1 · original submission · Major Revisions

This is an important and current field and I would like your paper to be published so I would ask you to look carefully at the feedback given from the longer review. Thsi raises a raft of very important issues that I am sure you can deal with systematically.

·

Basic reporting

An interesting article and topic, making use of a rare data set. I have left a number of suggestions of various level of significance (some quite strategic, some very specific) on the pdf of the manuscript. Please have a look and try and address them.

I feel overall the collected data has value. Before it's accepted for publication I would however suggest particularly re-visiting the following:
- giving the reader more confidence in how plant taxonomic data were collected? with species identification being outsourced - how can be confident this was done well?
- could some examples be provided of what actual species were identified in various categories (native/non native but also tree class sizes). This would have real value for a more biologically-inclined readers.
-both in the Introduction and Discussion providing a more detailed, deeper discussion of why is the nativeness (or otherwise) important in the context of this study? As it reads now, (to me at least) - the paper is an inventory of the things you found and catalogued. But it would be good to know 'so what'? Why is more native important (or not)? Does it have societal/ecological value or are some other aspects of what you found of more significance? More discussion of socioeconomic status vs what's found in gardens would also be good. There are some papers published in the last 2-3 years on a topic of SE status/ownership vs attitudes and plant choices which you could use to compare. Any specific species which are of particular value? Don't be afraid of making paper more plant-y/ecological!

Experimental design

I have no additional comment

Validity of the findings

Could another reviewer please look at Tables 2 & 3. I am not familiar with his type of analysis and can't really judge the validity/clarity of the information presented in them.

·

Basic reporting

There are a few cases of incorrect grammar and punctuation, noted in the attached marked pdf. As also noted there, I found the part of the Results section that described the separate analyses of tree size classes somewhat confusing.

Experimental design

This seems thorough and careful, though I am not qualified to evaluate the modeling aspects.

Validity of the findings

The manuscript uses a valuable and unusually thorough dataset to address some very interesting and important questions about trees planted by people in urban settings. As noted above, I do not consider myself qualified to evaluate the modeling methods and I thought one part of the Results a bit confusing, but overall I found the authors' analyses to be thorough and careful and their interpretations and conclusions to be sound and appropriate.

Additional comments

Please see comments in areas 1-3 above as well as comments and suggestions included in the attached marked pdf.

---

## Round 0.2 · Minor Revisions

Please ensure you discuss the issue of local source material or provide citations to better justify this. You have made substantial improvements to the manuscript but some areas still require a better explanation.

·

Basic reporting

Dear Wei and colleagues

While you have addressed some of the comments I made, I feel there are areas yet to be improved. This is particularly around the justification of the origin of species (especially down to a very 'cellular' level - native to city!!): why is this important and why should readers be interested in that? I am not disputing that this may be important, but as a reader I would benefit from you providing a more detailed context of why is that relevant/important/what to do with that knowledge?

I am copying from my previous review the two points which I would suggest you please address in more detail:

-both in the Introduction and Discussion providing a more detailed, deeper discussion of why is the nativeness (or otherwise) important in the context of this study? As it reads now, (to me at least) - the paper is an inventory of the things you found and catalogued. But it would be good to know 'so what'? Why is more native important (or not)? Does it have societal/ecological value or are some other aspects of what you found of more significance? More discussion of socioeconomic status vs what's found in gardens would also be good. There are some papers published in the last 2-3 years on a topic of SE status/ownership vs attitudes and plant choices which you could use to compare. Any specific species which are of particular value? Don't be afraid of making paper more plant-y/ecological!
- could some examples be provided of what actual species were identified in various categories (native/non native but also tree class sizes). This would have real value for a more biologically-inclined readers.

As I stated in the initial review, Table 2 in the paper is not discussed in Results or Discussion. What is the significance of it and what is it telling us? Please provide this information somewhere in the paper if you are keen to include this data.

Experimental design

no comment

Validity of the findings

no comment

·

Basic reporting

No comment.

Experimental design

No comment.

Validity of the findings

No comment.

Additional comments

Concerns raised in my first review have been satisfactorily addressed. Thank you.
- Dan Potter, UC Davis

---

## Round 0.3 · accepted · Accept

The edits now made make the manuscript acceptable for publication.

·

Basic reporting

See below

Experimental design

See below

Validity of the findings

See below

Additional comments

The revision is acceptable.